# Evaluating the Potential of Marine Invertebrate and Insect Protein Hydrolysates to Reduce Fetal Bovine Serum in Cell Culture Media for Cultivated Fish Production

**DOI:** 10.3390/biom12111697

**Published:** 2022-11-16

**Authors:** Inayat Batish, Mohammad Zarei, Nitin Nitin, Reza Ovissipour

**Affiliations:** 1Future Foods Lab and Cellular Agriculture Initiative, Virginia Seafood Agricultural Research and Extension Center, Virginia Polytechnic Institute and State University, Hampton, VA 23699, USA; 2Department of Food Science and Technology, Virginia Polytechnic Institute and State University, Blacksburg, VA 24061, USA; 3Department of Food Science and Technology, University of California-Davis, Davis, CA 95616, USA; 4Department of Biological Systems Engineering, Virginia Polytechnic Institute and State University, Blacksburg, VA 24060, USA

**Keywords:** serum-free media, cultivate meat, bioprocessing, protein hydrolysates

## Abstract

The use of fetal bovine serum (FBS) and the price of cell culture media are the key constraints for developing serum-free cost-effective media. This study aims to replace or reduce the typical 10% serum application in fish cell culture media by applying protein hydrolysates from insects and marine invertebrate species for the growth of Zebrafish embryonic stem cells (ESC) as the model organism. Protein hydrolysates were produced from black soldier flies (BSF), crickets, oysters, mussels, and lugworms with a high protein content, suitable functional properties, and adequate amino-acid composition, with the degree of hydrolysis from 18.24 to 33.52%. Protein hydrolysates at low concentrations from 0.001 to 0.1 mg/mL in combination with 1 and 2.5% serums significantly increased cell growth compared to the control groups (5 and 10% serums) (*p* < 0.05). All protein hydrolysates with concentrations of 1 and 10 mg/mL were found to be toxic to cells and significantly reduced cell growth and performance (*p* < 0.05). However, except for crickets, all the hydrolysates were able to restore or significantly increase cell growth and viability with 50% less serum at concentrations of 0.001, 0.01, and 0.1 mg/mL. Although cell growth was enhanced at lower concentrations of protein hydrolysates, the cell morphology was altered due to the lack of serum. The lactate dehydrogenase (LDH) activity results indicated that BSF and lugworm hydrolysates did not alter the cell membrane. In addition, light and fluorescence imaging revealed that the cell morphological features were comparable to those of the 10% serum control group. Overall, lugworm and BSF hydrolysates reduced the serum by up to 90% while preserving excellent cell health.

## 1. Introduction

The world population is projected to reach 10 billion by 2050, requiring a 70% increase in meat production to meet global demand [1]. Animal-based products, especially meats, make up a substantial portion of global protein consumption [2]. Among animal-based meat, aquatic food products constitute 20% of the total animal protein consumed currently, and this consumption has increased rapidly from previous years [1]. Conventionally, aquatic food products are obtained by fisheries or from different aquaculture systems, which are facing many challenges, including but not limited to overfishing, fraud, by-catch, antibiotic-resistant bacteria, emerging diseases, microplastics, and pollution [3,4]. Cultivated meat appears to be a sustainable technique compared to conventional meat, with many environmental, economic, and health benefits over other conventional alternatives [5]. Cultivated meat involves culturing cells or tissues in specifically formulated media that promote cell proliferation, metabolism, and differentiation [6]. Cell culture media accounts for 85–95% of the total cost of the cultivated meat process [7], and fetal bovine serum (FBS) is one of the most expensive components of cell culture media. FBS has a critical role in supporting cell proliferation and metabolism. Due to high costs, the risk of disease, high demand/low supply, high variability, the inability to grow specific cells, and ethical source issues [7], developing serum-free media for the cultivated meat industry could reduce the cost of the final products and address sustainability concerns. Despite successful efforts to develop serum-free media for medical applications, the high cost of the current serum-free media and non-food grade compounds in these media limits the suitability of these media for cultivated meat production. Developing serum-free media for cultivated beef has been thoroughly studied [7]. In addition, one of the critical challenges in food supply chains is food loss issues that present significant sustainability and security challenges, with 60 percent of meat becoming processed waste (1.4 billion tons for livestock; 800 million tons for seafood) [8,9]. Oysters and mussels could be applied for protein extraction/hydrolysis for feeding the cells in the cultivated meat industry without competing with the farmed bivalves industry. Lugworms are also other marine invertebrates that have not been studied before for enzymatic hydrolysis. Using insects as biorefinery for converting wastes to biopolymers is also a rapidly growing industry due to having fewer environmental issues compared to the conventional meat sources of protein, such as beef, chicken, and pork. However, one of the major issues is the limited market for insect products, despite the high quality and nutritional value of insect protein. Therefore, using insects as a source of protein to produce the protein hydrolysates/peptides to feed the cells and reduce the serum in cell-related media could be an efficient and alternative way to increase the application of insect protein and production of marine-based cultivated meat.

Peptides (protein hydrolysates) from different sources, including soy, have been used by many researchers for growing cells [10,11,12,13], mainly mammalian cells. However, to the best of our knowledge, no serum-free media are available for cultivated aquatic food products. Thus, developing a serum-free media using cost-effective sources of growth factors and hormones will be an efficient strategy to industrialize cultivated aquatic food products. Thus, developing a serum-free medium using cost-effective sources of growth factors and hormones will be an efficient strategy to industrialize cultivated aquatic food products. This study aimed to develop protein hydrolysates from two insect species, including the black soldier flies (BSF) and crickets, as well as three marine invertebrates, including oysters, mussels, and lugworms, for reducing serum in fish cell media. Moreover, we will evaluate the impact of different concentrations of protein hydrolysates on the doubling time, cell biomass, and cell performance. Extracellular lactate dehydrogenase leakage (LDH) was selected as a rapid measure of cellular integrity [14,15]. This measurement enables researchers to determine if a medium allows for the proper proliferation of cells and maintains cell health over extended periods without exerting any adverse effects.

## 2. Materials and Methods

### 2.1. Materials

The BSF was provided by Chapul Farm (McMinnville, OR, USA), and the cricket powder was obtained from Cricktone (Saint Louis, MO, USA). Oyster, mussel, and lugworm were provided from local stores. The enzyme used in this study was commercially available Alcalase^®^, an endoprotease enzyme (2.4 AU/g) from *Bacillus licheniformis* (Sigma–Aldrich Inc., St. Louis, MO, USA). The zebrafish embryonic stem cell line ZEM2S CRL-2147™ was obtained from The American Type Culture Collection (ATCC). The antibiotics which were initially used to cultivate zebrafish embryonic stem cells (ESCs) were obtained from Cytiva (Marlborough, MA, USA).

### 2.2. Protein Hydrolysates Production

The enzymatic hydrolysis process of proteins of various sources was performed according to previous studies [16], with slight modifications depending on the substrate (Figure 1A). The hydrolysates were prepared in at least six replicates based on the specific hydrolysis conditions (Table 1). Briefly, raw materials were mixed with water at a specific ratio (Table 1), heated to 60 °C for 20 min followed by adding the enzyme and hydrolysis for 1 h. The enzymatic reaction was terminated by heating the samples at 90 °C for 10 min. Samples were centrifuged, and protein hydrolysates were collected and freeze-dried. The dry yield (%) and productivity (mg/mL) were determined using Equations (1) and (2). Until further use, the lyophilized protein hydrolysates were kept at −20 °C.
(1)$$Yield\;\left(\%\right)=\frac{weight\;of\;dried\;protein\;hydrolysates\;\left(g\right)}{weight\;of\;raw\;materials\;\left(g\right)}\times 100$$
(2)$$Productivity\;\left(\frac{mg}{mL}\right)=\frac{Weight\;of\;freeze-dried\;protein\;hydrolysate\;\left(mg\right)}{Reaction\;volume\;\left(mL\right)}$$

### 2.3. Amino Acid Analysis, Protein Content, and Degree of Hydrolysis

The total amino acid analysis was determined according to the Association of Official Agricultural Chemists (AOAC) 982.30 E (a,b,c) [21]. Overall, after the samples’ digestion with 6N HCl, ion exchange chromatography was employed with post-column ninhydrin derivatization and quantitation. The crude protein content was assessed using AOAC standard method, Kjeldahl [21]. Briefly, the protein content was determined by multiplying the crude nitrogen content with nitrogen-to-protein conversion factor (*Kp*), including 4.43 for BSF [22], 5 for cricket [23], and 6.25 for all animal sources.

The protein quality assessment was conducted using the Digestible Indispensable Amino Acid Score (*DIAAS*) as recommended by the FAO [24] using Equation (3).
(3)$$DIAAS\;\left(\%\right)=100\times lowest\;value\;\left(\frac{mg\;of\;DIAA\;in\;1\;g\;of\;protein}{mg\;of\;DIAA\;in\;1\;g\;of\;reference\;protein}\right)$$

The degree of hydrolysis was measured using indirect formol titration [25].

### 2.4. Functional Properties

#### 2.4.1. Oil Holding Capacity (OHC)

The oil holding capacity was tested for all protein hydrolysates according to Shahidi, et al. [26] by mixing 500 mg of protein hydrolysate with 10 mL of pure canola oil. This mixture was left at room temperature for 30 min and gently mixed every 10 min, followed by centrifugation at 2500× *g* for 10 min at room temperature. The OHC was calculated according to Equation (4).
(4)$$Oil\;holding\;capacity=\frac{Mass\;of\;sample\;\left(with\;oil\right)-Mass\;of\;dry\;sample\;\left(g\right)}{Mass\;of\;dry\;sample\;\left(g\right)}$$

#### 2.4.2. Emulsifying Capacity (EC)

The EC was evaluated by mixing 500 mg of protein hydrolysates with 50 mL of 0.1 M sodium chloride solution in a 250 mL conical flask at room temperature, according to Yasumatsu, et al. [27]. The EC was calculated using Equation (5).
(5)$$Emulsifying\;capacity\;\left(\%\right)=\frac{Total\;volume-Aqueous\;volume}{Total\;volume}\times 100$$

#### 2.4.3. Foaming Capacity (FC)

The foaming capacity was calculated according to Pacheco-Aguilar, et al. [28] by mixing 750 mg of protein hydrolysates in 25 mL distilled water for 10 min, followed by homogenization for two min. The FC was calculated using Equation (6).
(6)$$Foaming\;capacity\;\left(\%\right)=\frac{Aerated\;volume-normal\;volume}{Normal\;volume}\times 100$$

### 2.5. Cell Culture and Maintenance

The Zebrafish embryonic stem cell (ZEMS2) was obtained from the American Type Cell Culture (ATCC). The media used to culture cells initially was Leibovitz-L-15 media (L-15), Dulbecco’s modified eagle media (DMEM), and Ham’s F12 media (F-12 media) in a ratio of 15:50:35, respectively, with buffering agents added (20 mM HEPES and 0.18 g/L sodium bicarbonate with 10% FBS). The cells in the ampoule were thawed at 28 °C and resuspended in 5% FBS in 9 mL media in T-25 cm^2^ for 30 min and then more FBS was added to make up 10% FBS. The subculturing process was conducted when cells reached 80–85% confluency by rinsing the cells with PBS and adding TrypLE Express to detach cells from the flask surface. The flasks were kept at 28 °C for 5 min to allow complete detachment, and then the cells were transferred to a 15 mL falcon tube. The TrypLE Express was neutralized by serum-free media (L-15 media) and centrifuged at 130× *g* for 8.5 min and a white pellet was obtained. The supernatant was removed, and the pellet was resuspended in 10% serum-containing media. The cell number and viability were analyzed using an automatic cell counter, and cell splitting was performed based on that. The cells reached confluency in a week, with media change once on the third day.

### 2.6. Cell Performance

The cells’ growth was studied daily for three days using a phase-contrast microscope equipped with an image analysis software. Direct image analysis is a dependable and accurate technique that has been demonstrated to be equivalent to other cell counting methods [29,30]. After 24 h of incubation in media containing 10% serum, the experimental media were changed to media containing 0, 1, 2.5, 5, and 10% serum.

Various protein hydrolysates were applied at different concentrations in three different experiments. In Experiment I, the impact of different concentrations of FBS on cell growth was evaluated. In Experiment II, various concentrations of protein hydrolysates (0.001, 0.01, 0.1, 1, and 10 mg/mL) in combination with media containing 0, 5, and 10% serum were studied. In experiment III, lower concentrations of protein hydrolysates (0.001, 0.01, and 0.1) (except that of crickets) in combination with media containing lower concentrations of serum (0, 1, 2.5, and 10%) were studied. The protein hydrolysate samples were prepared by mixing them in filtered, sterilized (0.22 µm) water. All the experiments were conducted in triplicate, with an initial cell density of 50,000 cells/mL. Cells were incubated for 24 h at 28 °C in a media with 10% serum to attach the plates. The cell numbers and morphology were recorded every 24 h and continued for three days using CKX-CCSW confluency checker by taking three images per well. All the hydrolysate conditions were tested in four biological replicates with three technical replicates each. The growth rate and doubling time were calculated according to Equations (7) and (8).
(7)$$growth\;rate\;\left(h^{-1}\right)=\frac{Ln\;\frac{\left(Viable\;cells\;at\;passaging\right)\;}{viable\;cells\;at\;seeding}}{Time\;\left(h\right)}$$
(8)$$doubling\;time\;\left(h\right)=\frac{Ln\;\left(2\right)}{growth\;rate\;\left(h^{-1}\right)}$$

The viability of cells was tested by the PrestoBlue staining method. After 72 h, the reagent 100 µL of PrestoBlue was added to the wells and incubated at 28 °C for 2 h, followed by measuring the absorbances at 570 and 600 nm using a microplate reader. The dye reduction was calculated according to Equation (9).
(9)$$Reagent\;reduction\;\left(\%\right)=\frac{\left(117216\times A1\right)-\left(80586\times A2\right)}{\left(155677\times N2\right)-\left(14652\times N1\right)}\times 100$$
where *A1*, *A2*, *N1*, and *N2* are the absorbance of sample wells at 570 and 600 nm and media.

### 2.7. Fluorescent Imaging

Optimum concentrations of protein hydrolysates were selected for imaging. The cells were fixed for 10 min with 4% paraformaldehyde and rinsed twice with PBS to remove the paraformaldehyde. The cells were made permeable to dyes by incubating them for ten min with 0.1% Tween, followed by rinsing with PBS. In order to visualize cell nuclei, 25 µL of Hoechst dye was diluted with PBS at a ratio of 1:2000, added to the cells, and incubated in the dark for 10 min. After incubation, the cells were washed twice with PBS, and then 1 mL of PBS was added to each well prior to the addition of the second dye for cytoskeleton visualization. Each well was given two drops of actin green dye, incubated for 30 min, and then washed twice with PBS. The cells were then suspended in a live cell imaging solution to improve the image quality. The cells were subsequently observed using a fluorescent microscope with UV excitation/emission wavelengths of 361/486 nm and blue-cyan light with excitation/emission wavelengths of 495/518 nm.

### 2.8. Lactate Dehydrogenase Activity (LDH)

The lactate dehydrogenase activity of the most potent protein hydrolysate concentrations was evaluated for 2.5 and 1% serum concentrations (only in Experiment II). Cells were cultured for three days under optimum conditions. After three days, the supernatant was harvested, then transferred to 96 well plates, and the test was conducted following the instructions provided by the kit. Four biological replicates and three technical replicates per biological replicate were used for statistical reproducibility.

## 3. Statistical Analysis

The statistical analysis was conducted using JMP v.16 software. The data were tested for normality and homoscedasticity to confirm normal distribution. Normality was tested using the Shapiro–Wilk test and normal quantile plots (NQP), and the even distribution of variability was tested using Levene’s test. If the data were found to be normally distributed, a one-way ANOVA and Tukey’s HSD (honestly significant difference) test was performed; otherwise, the Wilcoxson/Kruskal–Wallis test and Steel Dwass/Wilcoxson pair comparisons were performed with their respective controls. Differences of *p* < 0.05 were considered statistically significant.

## 4. Results and Discussion

### 4.1. Protein Quality

The amino acid composition, protein content, and protein quality of the BSF, cricket, oyster, lugworm, and mussel hydrolysates are presented in Table 2. As shown, the protein content of cricket and BSF hydrolysates is higher than that of lugworm, mussel, and oyster hydrolysates. All protein hydrolysates contained a high concentration of essential amino acids with cell-growth-promoting properties, such as alanine, glycine, proline, and aspartic acid. Protein hydrolysates were rich in glutamic acid, which also plays an essential role in animal cell culture [31,32]. Glutamic acid is an important source of nitrogen for de novo amino acid synthesis, which contributes to protein and nucleic acid production. In addition to providing skeletons for carbon and nitrogen biosynthesis, glutamic acid is a significant replenisher of metabolites in the tricarboxylic acid (TCA) cycle [33]. Hydrophobic amino acids, such as glycine, alanine, valine, and leucine, were also dominant amino acids, suggesting the presence of bioactive peptides that can have potential growth-augmenting activities. All protein hydrolysates showed a high protein quality measured based on the Digestible Indispensable Amino Acid Score (DIAAS) content of essential amino acids. BSF showed the highest DIAAS score, followed by lugworm and then cricket. The lowest DIAAS score was related to mussel protein hydrolysates and then oyster. Although the cricket protein hydrolysates were revealed to have the highest protein content, their DIAAS score was not as high as that of BSF due to the difference in their amino acid profile and composition.

### 4.2. DH and Techno-Functional Properties

The results of DH and functional properties are presented in Figure 1A–D. Marine invertebrates (mussels, oysters, and lugworms) showed significantly higher DH compared to the insects (BSF and cricket) (*p* < 0.05). DH can influence the number of peptides released and their size, conformation, and amino acid sequence, which imparts various functional and bioactive properties to the hydrolysate product [17,34].

In this study, insect protein hydrolysates showed OHC values ranging between 3.54 and 4.83 g/g, which was within the previously reported range, including BSF (0.8–5.2 g/g) [35,36,37] and cricket (1.42–3.5 g/g) [38,39,40]. Crickets had the highest OHC compared to the other sources, and the mussels and oysters were revealed to have the lowest OHC among the selected protein sources (*p* < 0.05), which could be due to the higher content of hydrophobic amino acids in oyster, mussel, and lugworm hydrolysates compared to those of BSF and cricket hydrolysates. To the best of our knowledge, this is the first report of a lugworm protein hydrolysate OHC that is close to that of insect protein hydrolysates.

The higher OHC of the insect protein hydrolysate is potentially related to the higher content of hydrophobic amino acids in insect hydrolysates. Oil-holding capacity potentially can be tied to various positive attributes of protein hydrolysates that can be useful in media formulations for animal cell cultures. The culture medium comprises several hydrophobic components, such as insulin, growth factors, hormones, and carrier proteins, which are less soluble but essential for cell growth, survival, and proliferation [6]. Due to their rigid structure and non-polar chains, lipid-based components, such as sterols, cholesterol, and fatty acids, are difficult to solubilize in cell media. Lipids are essential micronutrients for cell proliferation, as these are required for membranes and play a role in signal transduction. Lipids are supplied exogenously, as many cells cannot produce them [41,42]. Hydrolyzing proteins expose their network and hydrophobic amino acids, which aid in trapping oil and potentially oil-based components, thereby boosting OHC and aiding in the solubilization and stabilization of these oil-based components [43]. In addition, a high oil-holding capacity is associated with increased hydrophobicity and hydrophobic amino acids [44].

Mussel, oyster, and lugworm protein hydrolysates had the highest foam capacities compared to that of insects (*p* < 0.05) (Figure 1B). Foaming capacity is a significant function feature for many food-based products, adding to the texture or appearance. For example, bubbles define the texture of aerated commercial products, including bread, ice cream, mousse, and meringue. [45]. Foaming in a bioreactor during processing is a typical occurrence. However, it can cause significant issues, such as inhibiting cell development by limiting the surface area contact between the growth media and the bioreactor headspace and decreasing the oxygen transfer rates. In addition, foam accumulation in excess can clog filters and cause vessel overpressures that exceed allowable limits. As a result, hydrolysates should not increase the foaming that occurs spontaneously in a bioreactor when added to media. The insect hydrolysates produced in this study had foam capacities of 4 and 20%, which are extremely low compared to those of the cricket and BSF found in the literature, which were 39–100% [17,35]. The bivalve protein hydrolysates showed a lower foaming capacity (40–50%) than snails and scallops (51.1–160%) [46,47]. Foaming qualities are generally imparted by partially denaturing a globular protein as hydrophobic regions are exposed. These regions can efficiently adsorb into air–water interfaces and lower interfacial tension, thus enhancing foaming capability. However, extensive denaturation will reduce the proteins’ ability to produce foams [48].

The results of emulsion capacity indicated that lugworm protein hydrolysates had the lowest emulsion capacity (*p* < 0.05). Insect hydrolysates’ emulsion capacity was in the range of 57.8–60%, which is in agreement with other researchers’ findings (39–98%) [35,49,50]. The emulsion capacity of bivalve mollusks fell consistently in the 35–65% range as found by other studies [51,52]. An increase in emulsion capacity is usually related to the amphipathic nature of peptides that are produced by hydrolysis. It is widely reported that peptides higher than the size of 2 kDa have a higher emulsion capacity as they may possess both hydrophobic and hydrophilic amino acid chains that can interact with both water- and oil-based components. However, the lower emulsion capacity can be attributed to the production of extremely short peptides, which reduce the overall amphipathic nature of the hydrolysates. Emulsion capacity is also another critical functional property that has industrial applications when it comes to cultivated meat production. Culturing cells requires many water- and oil-based components to ensure proper cell growth, proliferation, and sustenance; however, the presence of these opposite groups makes the culture media thermodynamically unstable [53]. Emulsifying various components can help solubilize these opposite groups and allow their transfer to the cells when required. The hydrolysis of proteins exposes many amphipathic, polar, and non-polar side chains that can help solubilize oil-based components and create stable emulsions so that all the necessary components can be utilized by cells when required [54].

### 4.3. Cell Morphology, Growth, and Viability

This study aimed to apply whole-protein hydrolysates from different sources to reduce or replace serum only by applying protein hydrolysates without adding other growth factors.

### 4.4. Impact of Serum Concentration

For zebrafish ESCs, 10% serum is the standard required serum concentration for regular cell growth. In the first experiment, different serum concentrations (0, 1, 2.5, 5, and 10%) were evaluated on the cell growth and doubling time. The impact of the serum at various concentrations on many parameters of cell growth and viability is demonstrated in Figure 2.

As the serum concentration decreases, the cell density reduces significantly (*p* < 0.05) (Figure 2). By decreasing the serum from 10% to 5, 2.5, and 1%, while the cell numbers are reduced by less than 0.2 log, the cell morphology did not change, demonstrating that although the cells are affected by the serum reduction, they retain their original morphology (Figure 2A). At a serum concentration of 0%, there was a significant cell reduction in the rate of cell growth and morphological changes with an increase in rounded cells. This shows that although cell growth and proliferation decreased at a 2.5% serum concentration, this amount of serum was enough to maintain the normal morphology of cells. At 1% serum, the cells started showing some morphological aberrations, and severe morphological anomalies were observed with complete serum starvation. Overall, as the serum reduced, the cell death increased as the cells lost their original spindle-like fibroblast morphology, and became rounded, more elongated, spiky, and starved in nature, which was also observed by other researchers [55]. The doubling time was increased from 14.6 h at 10% serum [56,57,58] to 15.54, 16.37, and 18.5 h at 5, 2.5, and 1% serum, respectively. The doubling time was increased to 24.5 h in serum-free media (Figure 2C).

In Experiment II, we evaluated the impact of different protein hydrolysate concentrations from 0.001 to 10 mg/mL in combination with different concentrations of serum (0, 5, and 10%) on cell performance. The cell growth and cell viability results in media containing different concentrations of protein hydrolysate are presented in Figure 3 (1 and 10 mg/mL) and Figure 4 (0.001, 0.01, and 0.1 mg/mL).

Overall, providing 1 and 10 mg/mL protein hydrolysates significantly reduced cell growth and cell viability (*p* < 0.05). Mussel protein hydrolysates at 10 mg/mL concentrations significantly reduced cell growth compared to other hydrolysates (*p* < 0.05). Meanwhile, at 1 mg/mL concentrations, no significant differences were observed among the hydrolysates in terms of cell growth (*p* < 0.05).

**Figure 3 biomolecules-12-01697-f003:**
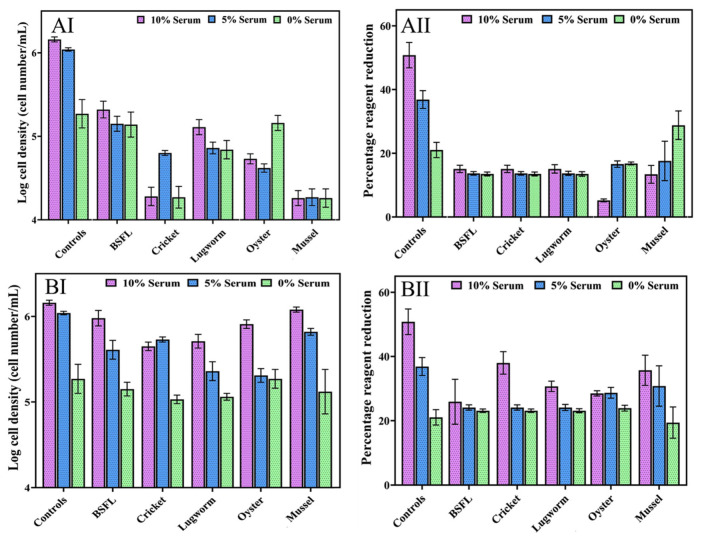
ZEM2S cells’ growth in media containing different serum concentrations (10%, 5% and 0%), (**A**) 10 mg/mL protein hydrolysates (AI: cell growth; AII: cell viability) and (**B**) 1 mg/mL protein hydrolysate (BI: cell growth; BII: cell viability).

In addition, within 24 h of adding a medium containing protein hydrolysates, the cells lost their fibroblast-like morphology, and cell death was induced. Adding these protein hydrolysate concentrations decreased the media pH despite a buffering mechanism, as the oxidation of phenol red caused the yellowing of the medium.

**Figure 4 biomolecules-12-01697-f004:**
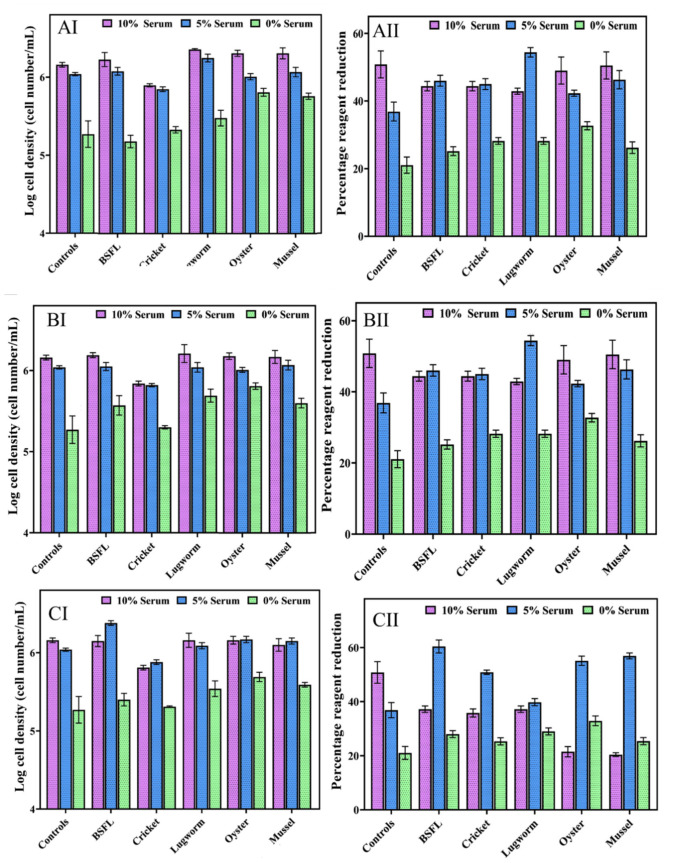
ZEM2S cells’ growth in media containing different serum concentrations (10%, 5% and 0%), (**A**) 0.1 mg/mL protein hydrolysates (AI: cell growth; AII: cell viability); (**B**) 0.01 mg/mL protein hydrolysate (BI: cell growth; BII: cell viability); and (**C**) 0.001 mg/mL protein hydrolysates (CI: cell growth; CII: cell viability).

The lower concentrations of the lugworm and mussel (0.01 and 0.1 mg/mL) in combination with 10% FBS exhibited a significantly higher cell proliferation compared to that of the control group with 10% FBS. The lowest concentration of protein hydrolysates (0.001 mg/mL) in combination with 5% serum resulted in even higher cell growth than the other tested concentrations (0.01–10 mg/mL).

At a 5% serum concentration in combination with 0.01 and 0.1 mg/mL protein hydrolysates, only mussel and lugworm hydrolysates demonstrated significantly higher growth than the 10% control. Meanwhile, at the lowest concentration of protein hydrolysates (0.001 mg/mL), BSF, and oyster hydrolysates, there was significantly increased cell growth compared to the 10% serum control group, demonstrating that depending on the type of protein hydrolysates and concentrations, supplementing the media with protein hydrolysates can reduce the serum by up to 50%. Using a low concentration of protein hydrolysates from lugworm, oyster, mussel, and BSF improved the cell growth significantly in a serum-free medium.

As depicted in Figure 5, at low concentrations of protein hydrolysates (0.001, 0.01, and 0.1 mg/mL) in a serum-free medium, none of the hydrolysates were able to increase the cell growth parameters compared to the 10% serum control. However, except for the cricket hydrolysates, all the other hydrolysates were able to significantly increase cell growth compared to the serum-free media without protein hydrolysates (control), indicating that some protein hydrolysates could potentially support fish cell growth in a serum-free environment. While protein hydrolysates supported cell growth in serum-free media, the loss of cell morphology and starvation indicated that one source of protein hydrolysates alone could not maintain cell health, and a combination of different hydrolysates might be necessary (Figure 5).

These results agree with other researchers’ findings in which a high concentration (4 to 6 mg/mL) of protein hydrolysates, including soy [59] and fish gelatin [60], significantly decreased cell proliferation. In addition, wheat gluten hydrolysate at concentrations between 6 and 12 mg/mL reduced cell proliferation [61]. Protein hydrolysates derived from various wastes, such as eggshells and carcasses, at 10 mg/mL had a similar dose-dependent, cytotoxic effect on bovine stem cells [13]. This can be related to the presence of a high concentration of nutrients, such as oligopeptides, which disrupt the overall nutritional balance of the medium [60,62] and thus drastically alter its pH. In this study, the growth and population doubling time were negatively impacted as a result of the death of the cells, possibly by affecting the nutrient balance due to high amino-acid or oligopeptide concentrations, as reported by [60,62]. Lower concentrations of whey protein hydrolysate (0.01–0.5 mg/mL) boosted osteoblastic cell line proliferation, viability, and alkaline phosphatase activity [63]. Similar results were observed on bovine cells with algal extract and various other industrial byproduct hydrolysates; the cell growth increased relative to the serum-free conditions, albeit less than in the serum-rich conditions [13,64].

The degree of hydrolysis in this study ranged from 20 to 32%. It has been shown that cell growth in the presence of protein hydrolysates with different concentrations of serum depends on the degree of hydrolysis [65]. However, in this study, no correlation was observed between the degree of hydrolysis and cell growth. This could be explained by the fact that in this study enzymatic hydrolysis was conducted under one specific condition.

Protein hydrolysates have been demonstrated to have more than a nutritional effect on cells due to the peptides’ size, amino acid composition, and peptide amino acid sequences resulting in protective, anti-apoptotic, and growth-promoting properties. A study on CHO cell lines using several plant-based peptides revealed that while protein hydrolysates did not enhance the nutritional value of the basal media, they significantly boosted cell growth [66]. Similar effects were reported when CHO cells were grown with 10% serum and algal extracts [67].

In general, all the hydrolysates, except the cricket hydrolysate, were able to reduce the serum concentration effectively and, in some cases, the cell growth was higher than the control group containing 10% serum, demonstrating the tremendous potential of these protein hydrolysates to develop serum-free media for cellular agriculture. The concentrations of 1 and 10 mg/mL appear to have a negative effect on cell growth and survival and to be possibly toxic to cell growth overall. Protein hydrolysates at lower concentrations (0.001, 0.01, and 0.1 mg/mL), did not show toxicity. Overall, the cell viability determined using PrestoBlue was comparable to the results obtained by the microscope image analysis, which presents a good cross-reference test. The highest cell viability was related to BSL with a 65% reduction. 

**Figure 5 biomolecules-12-01697-f005:**
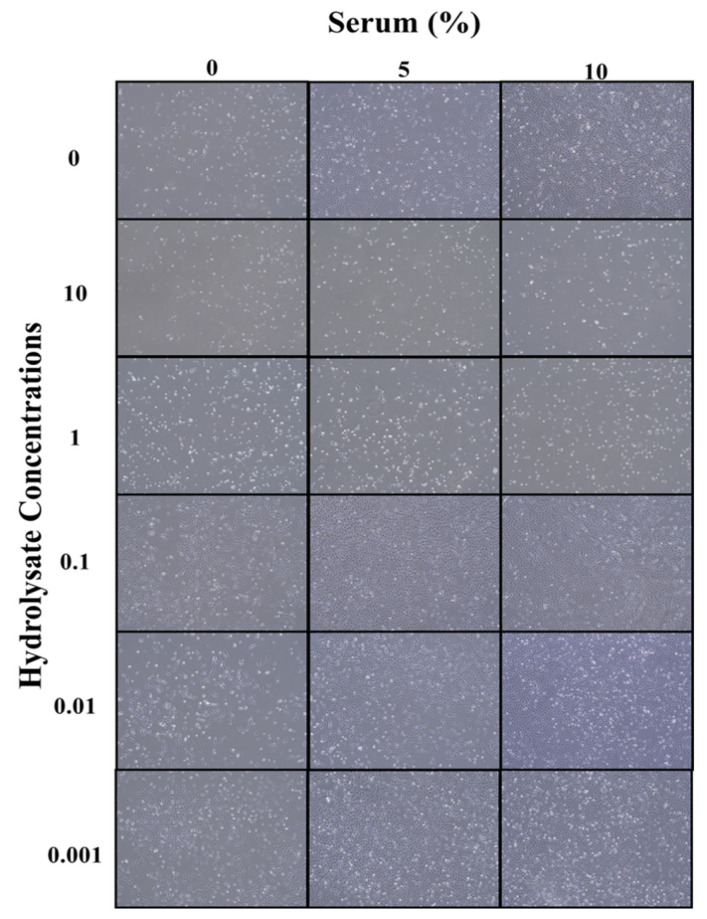
ZEM2S cells’ morphological changes in different media containing BSF hydrolysates (0–10 mg/mL) in combination with 0, 5, and 10% serum. The morphology was provided using phase-contrast microscope (200 µm).

In Experiment III, the effects of lower concentrations of protein hydrolysates (0.001, 0.01, and 0.1 mg/mL) except for cricket, in combination with lower concentrations of serum (1 and 2.5%) on cell performance were studied (Figure 6).

All the hydrolysates at lower concentrations illustrated a positive impact on cell growth in combination with 1 and 2.5% serum (Figure 6). Combining 0.001 mg/mL of protein hydrolysates with 1% serum increased cell biomass compared to the control groups containing 1, 2.5, and 10% serum, indicating that supplying protein hydrolysates at low concentrations can replace serum applications in cell culture media.

**Figure 6 biomolecules-12-01697-f006:**
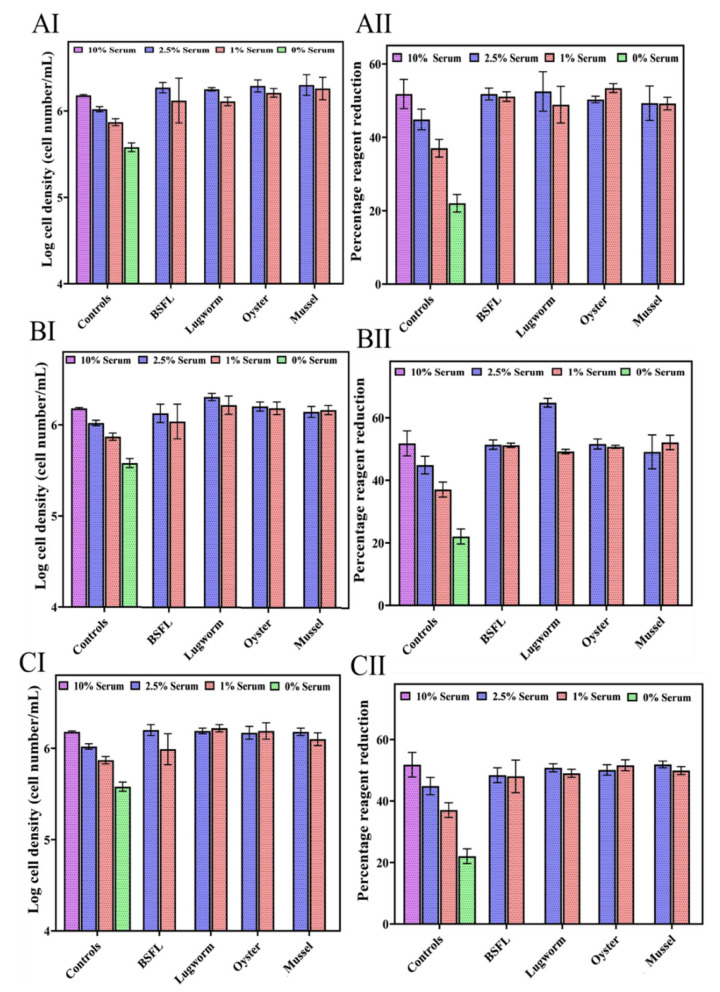
ZEM2S cells’ growth in media containing different serum concentrations (10%, 5% and 0%), (**A**) 0.1 mg/mL protein hydrolysates (AI: cell growth; AII: cell viability); (**B**) 0.01 mg/mL protein hydrolysate (BI: cell growth; BII: cell viability); and (**C**) 0.001 mg/mL protein hydrolysates (CI: cell growth; CII: cell viability).

Most protein hydrolysates exhibited high cell growth at 2.5%, as evidenced by their growth rates and population doubling times. The cell viability profiles of all protein hydrolysates did not differ significantly from the 10% standard. Compared to the 10% serum standard, lugworm and mussel hydrolysates displayed significantly higher growth. This suggests that protein hydrolysates may synergistically affect cell growth in combination with low serum concentrations. The 0.1 mg/mL concentration efficiently reduced serum levels by up to 75% without affecting the cell viability or growth profile. At 1% serum, only the mussel hydrolysate exhibited significantly more growth than the control at 10% serum.

Using 0.001 mg/mL protein hydrolysates indicated that only BSF hydrolysates significantly improved the cell proliferation compared to the 10% serum control group, while the other hydrolysates resulted in similar or slightly better cell proliferations compared to the 10% serum control group. Protein hydrolysate concentrations of 0.01 and 0.001 demonstrated statistically better or similar levels of growth and viability compared to those of the 10% control, showing that the serum concentration can be decreased by 75 to 90% without impacting the cell performance using protein hydrolysates. Morphologically, there was no discernible difference between the protein hydrolysates at any of the serum concentrations and 10% control. This demonstrates that protein hydrolysates between 0.001 and 0.1 mg/mL can restore cell growth, viability parameters, and morphological integrity.

### 4.5. Fluorescent Staining of Cells

Hoechst dye is an effective and reliable fluorophore with a long history of visualizing DNA content and nuclear structure using a fluorescence microscope. It is a fluorescent dye that can stain both living and fixed cells and emits fluorescence in the blue light spectrum. Serum deprivation induces apoptosis or necrosis, which can be recognized by karyopyknosis and visually identified by staining cells with Hoescht dye [68,69]. Actin is another ubiquitous protein in animal cells, particularly in the filamentous (F) form. These proteins maintain cellular shape, structure, signaling, and cell division [70]. It is known that serum deprivation impairs actin protein, which can potentially negatively impact essential cell-based features, such as cell shape, cell–cell matrix connections, and proliferative capacity [71]. The process of serum deprivation is poorly understood, as it differs from cell type to cell type. For this study, cells grown at all serum conditions (0–10%) and all protein hydrolysates selected from the previous section were stained with Hoescht and actin green (Figure 7).

From the fluorescent staining, it is evident that serum deprivation had an impact on the cytoskeleton, although there was no discernible effect on the nucleus other than a reduction in numbers. Using the 10% serum as the control, a high number of nuclei and abundant actin protein stains were observed. At a serum concentration of 2.5%, there was no visible effect on the nucleus other than a reduction in the number. However, the actin filaments were stained less, indicating a decrease in the presence of actin proteins, and the cell area was increased. As the serum content decreased from 1 to 0%, the actin protein exhibited minimum actin staining, and the cells were more dispersed and obtained a greater surface area. Overall, the actin protein presumably decreased along with the serum concentration, which altered the cell shape. This result is supported by a second investigation demonstrating a decrease in actin protein staining under serum-starvation conditions [71]. The selected protein hydrolysates from the first section exhibited the same number of nuclei and cytoskeleton as the 10% serum control, demonstrating that the protein hydrolysates could restore actin protein in these cells and, consequently, the original cytoskeleton.

### 4.6. LDH Staining

The LDH activity of the selected protein hydrolysates at 2.5 and 1% serum concentrations and with 10 and 0% controls is presented in Figure 8.

In general, a serum concentration of 0% showed a more significant release of LDH, indicating damage to the plasma membrane. In contrast, a serum concentration of 10% had the lowest LDH activity, and in some cases no LDH activity was observed. These results suggest that the integrity of the cell membrane was compromised, resulting in the release of LDH, which can indicate apoptosis and necrosis. Overall, the results were highly variable due to the serum’s inherent LDH activity, which can interfere with the kit. To overcome this challenge, for every study we collected specific blanks to minimize the interference from the serum.

All other groups had significantly greater LDH release than the 10% control group. Only BSL and lugworm hydrolysates significantly reduced the LDH release relative to the 0% control group. All other hydrolysates exhibited a higher LDH release, indicating that these protein hydrolysates can cause a greater degree of cell membrane damage than in serum-free conditions [14].

### 4.7. Yield and Cost of Protein Hydrolysates for Cultivated Meat Production

The yield, productivity, and cost of production of the protein hydrolysates are provided in Table 3. For production of a hydrolysate that properly supports animal cell culture growth, the whole organism was used instead of protein isolate or concentrate. The productivity and yield of protein hydrolysates were examined to determine their potential for large-scale manufacturing. The yield and productivity values for BSF, cricket, and mussel hydrolysates were higher or comparable to those found in the literature. The costs of production for 1 kg BSF, crickets, mussels, oysters, and lugworms were USD 91.51, USD 494.5, USD 570, USD 950, and USD 912. Based on the results of the cell growth, doubling time, and cell viability, the BSF and lugworms provided promising sources of protein hydrolysates for cell growth. According to this study’s results, the concentration of protein hydrolysate required for cultivated fish production is 0.001–0.1 mg/mL, and the yield and productivity appear suitable for cultivated meat scale-up. Culturing meat requires the cultivation of billions of cells using limited space, time, and resources. A two kg batch of any protein hydrolysate would be sufficient to supply the maximum capacity of a stirred bioreactor used for animal cell culture with a capacity of 2000 L to produce 10 to 100 kg of cultivated meat [72]. Based on the economic analysis, BSF provides a more sustainable and cost-effective source of protein hydrolysates for the cultivated meat industry. First of all, BSF feeds on agri-food wastes with the conversion ratio of 2 (2 kg waste to 1 kg BSF), generates low-greenhouse-gas emissions, produces high-protein-content materials, and has a high nutritional value [16].

## 5. Conclusions

In this study, the impact of different concentrations of protein hydrolysates in combination with various concentrations of serum on zebrafish embryonic stem cell performance was studied. All hydrolysates demonstrated the ability to replace at least 90% of the serum in cell culture conditions, reducing the overall cost of producing cell-based meat. Although protein hydrolysates could potentially be used for developing serum-free media, more investigations are required to address the morphological changes in the cells. Based on the results of this study, the BSL hydrolysates provided optimum DH, amino-acid composition, peptide size, and suitable functional properties for reduced serum or serum-free media development. At 2.5 and 1% serum concentrations, nearly all protein hydrolysates demonstrated excellent cell growth characteristics and viability at 0.001–0.1 mg/mL concentrations, indicating that protein hydrolysates should be applied at low concentrations to support cell growth. Additionally, fluorescence imaging has shown that protein hydrolysates can improve cytoskeleton density as high as the media containing the 10% serum as the control group. Lugworm and BSL hydrolysates exhibited the lowest LDH activity, indicating the least damage to the cell membrane, making them the optimum and most effective protein hydrolysates for serum-free media development without compromising cell health indicators. However, due to the cost of the protein hydrolysates, BSL provides the most cost-effective and sustainable source of protein hydrolysates for cell cultures. Further investigations are required to determine the most effective peptides’ molecular weight and sequence on cell growth. In addition, other commercially available enzymes and fermentation methods should be evaluated.

## Figures and Tables

**Figure 1 biomolecules-12-01697-f001:**
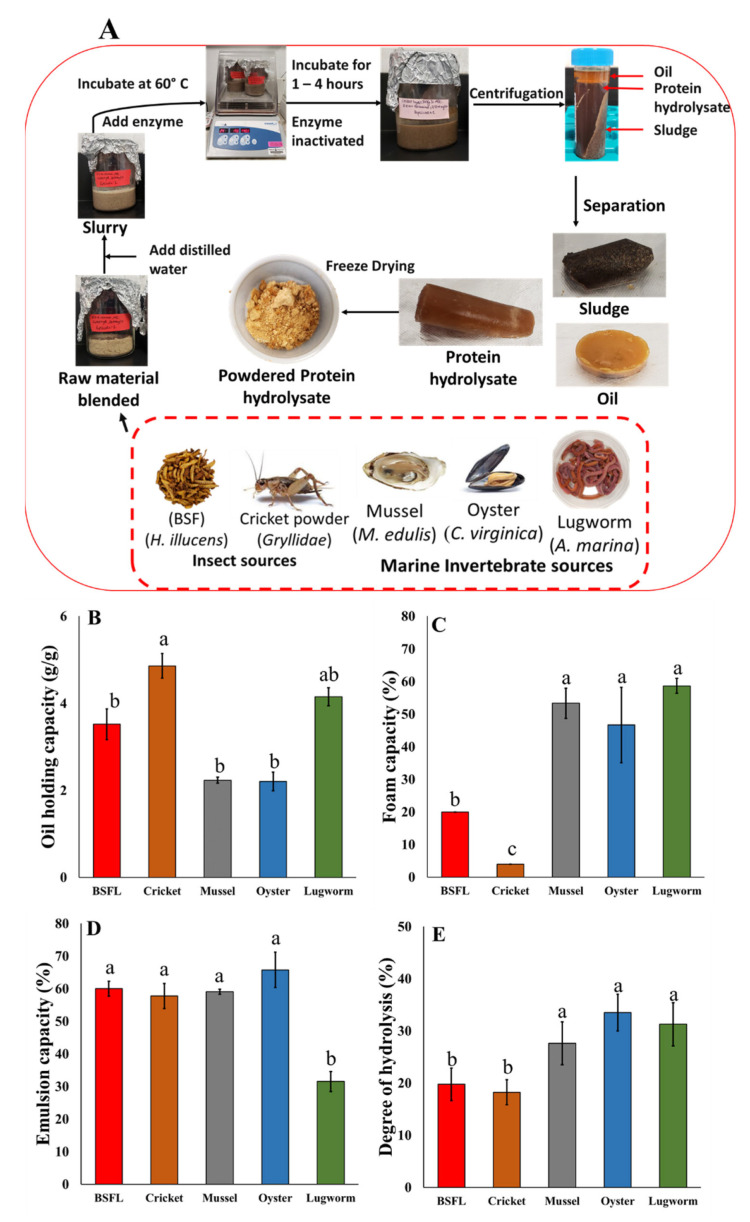
(**A**) Protein hydrolysates bioprocessing via enzymatic reaction; Techno-functional properties of protein hydrolysates; (**B**) oil-holding capacity; (**C**) emulsifying capacity; (**D**) foaming capacity; and (**E**) degree of hydrolysis.

**Figure 2 biomolecules-12-01697-f002:**
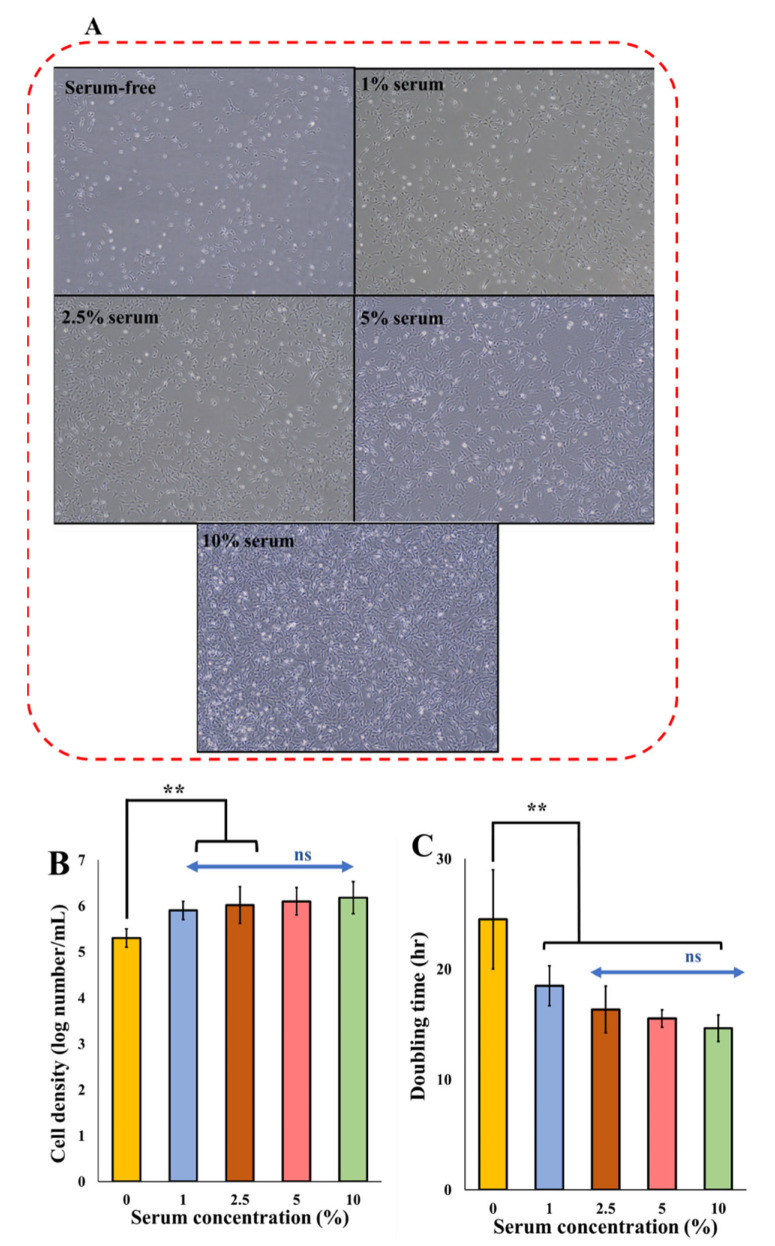
Cell growth parameters of ZEM2S cells at various serum concentrations: (**A**) The effect of different concentrations of serum on cell morphology and density through phase-contrast microscope; (**B**) Cell density at different concentrations of serum. ns: non-significant, (**) significant (*p* < 0.05); (**C**) doubling time calculated based on the specific growth rate of the cells. Micrographs of ZEM2S cells were obtained at 10× (200 µm) magnification.

**Figure 7 biomolecules-12-01697-f007:**
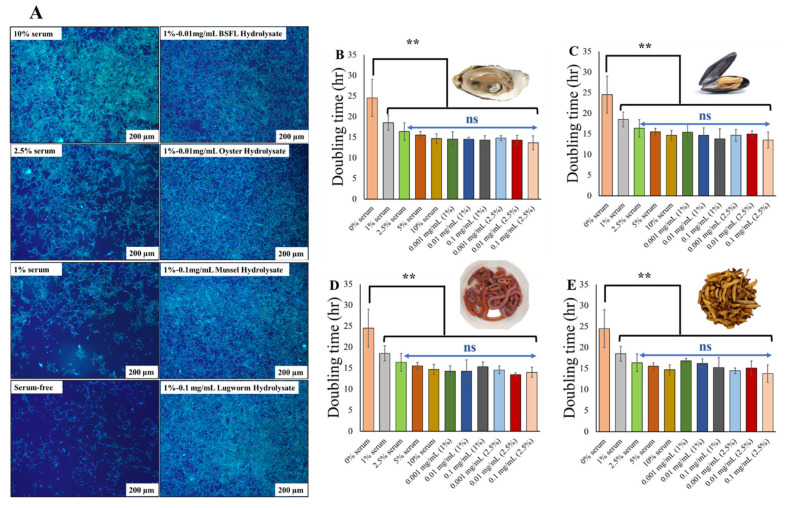
(**A**) Hoechst and actin green fluorescent staining of cells at all serum concentrations and protein hydrolysate conditions. Only selected concentrations of protein hydrolysates in combination with 1% serum were used for imaging; Doubling time (h) for protein hydrolysates at low concentrations (0.001, 0.01, and 0.1 mg/mL) in combination with low serum concentrations (1 and 2.5%), negative control (serum-free) and positive control (10% serum), for oyster (**B**); mussel (**C**); lugworm (**D**); and BSF (**E**). ns: non-significant; (**) significant (*p <* 0.05).

**Figure 8 biomolecules-12-01697-f008:**
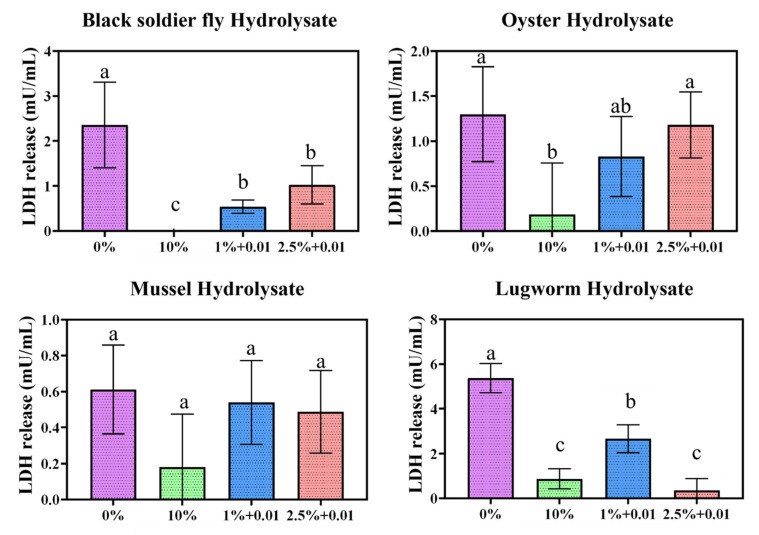
Lactate dehydrogenase assay conducted for protein hydrolysate conditions at 2.5 and 1% serum concentrations. a represents significantly different (*p* < 0.05) from 10% control, b represents significantly different (*p <* 0.05) from 0% control.

**Table 1 biomolecules-12-01697-t001:** Specific reaction conditions for enzymatic hydrolysis of substrates.

Substrate	Water to Substrate Ratio (*v*/*v*)	Enzyme (%)	Hydrolysis Time (h)	Reference
BSF	3:1	2	1	[16]
Cricket	3:1	2	1	[17]
Oyster	3:1	1	1	[18]
Mussel	3:1	2	1	[19]
Lugworm	1:1	2	1	[20]

**Table 2 biomolecules-12-01697-t002:** Amino acid content, protein content, and quality and degree of hydrolysis of all hydrolysates.

Amino Acid (g/100g of Protein)	BSL	Cricket	Oyster	Mussel	Lugworm
Phenylalanine	1.99	2.12	1.51	1.32	1.99
Valine	4.02	4.03	2.21	2.02	2.55
Threonine	2.41	2.91	1.91	2.04	2.16
Tryptophan	0.79	0.57	0.36	0.43	0.50
Methionine	0.85	1.25	1.07	0.88	1.13
Leucine	4.09	5.16	2.91	2.77	3.66
Isoleucine	2.72	2.84	1.98	1.84	2.34
Lysine	4.85	5.27	3.11	3.03	4.02
Histidine	1.85	1.69	0.83	0.87	1.14
Taurine	0.10	1.20	0.87	3.50	0.78
Hydroxyproline	0.00	0.20	0.58	0.28	0.31
Aspartic Acid	5.51	8.01	3.95	4.43	5.05
Serine	2.64	3.81	1.79	1.94	1.90
Glutamic Acid	8.51	10.52	4.81	5.95	6.71
Proline	3.94	3.98	1.60	1.77	2.06
Lanthionine	0.22	0.20	0.15	0.13	0.22
Glycine	3.23	4.33	2.09	3.77	2.70
Alanine	4.43	5.87	2.25	2.85	4.62
Cysteine	0.49	0.59	0.58	0.61	0.66
Tyrosine	3.53	2.65	1.68	1.48	1.61
Hydroxylysine	0.04	0.04	0.03	0.09	0.00
Ornithine	0.08	0.16	0.60	0.11	0.22
Arginine	3.47	5.76	2.91	2.82	3.51
Total amino acid content (g/100 g protein)	59.70	73.12	39.70	44.88	49.80
Protein content (%)	43.95	69.55	44.69	54.25	56.59
DIAAS score (%)	140.84	96.42	92.31	80.18	97.85
Protein quality	Excellent	Good	Good	Good	Good

**Table 3 biomolecules-12-01697-t003:** Dry yield and productivity (mg/mL) obtained for all protein hydrolysates.

Substrate	Dry Yield (%)	Productivity (mg/mL)	Yield and Productivity from the Literature ^1^	Raw Material Cost (per kg)	Cost/1 kg Protein Hydrolysates (USD) *	Cost for 100 kg Cultivated Meat (USD) **
BSF	16.57 ± 0.46	60 ± 0.00	10.7–6.9% 21 mg/mL 12.1–12.4%	15.16	91.51	0.915
Cricket	16.68 ± 1.45	60 ± 0.01	9.7–12.1% 5.7–11.2%	82.48	494.5	4.94
Mussel	9.78 ± 0.40	30 ± 0.00	5.27–8.66% 8.34%	55.75	570.1	5.70
Oyster	9.39 ± 0.95	30 ± 0.00	–	89.2	950	9.50
Lugworm	30.05 ± 1.24	100.16 ± 0.00	–	274.3	912.77	9.12

^1^ BSF: Abduh, et al. [73]; Firmansyah and Abduh [74]; Cricket Hall, Jones, O’Haire and Liceaga [17]; Trinh and Supawong [50]; Mussel: Mohd Rodzi [75]; Normah and Asmah [76]; Oyster and lugworm: not found; (*) Cost was calculated based on the yield and raw materials’ prices; (**) Cost of required protein hydrolysates for producing 100 kg cultivated meat in a 2000 L reactor.

## Data Availability

Not applicable.

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
