# Peer review of "Evaluating the Potential of Marine Invertebrate and Insect Protein Hydrolysates to Reduce Fetal Bovine Serum in Cell Culture Media for Cultivated Fish Production"

_biomolecules, 2022, doi:10.3390/biom12111697_

Round 1
Reviewer 1 Report
Interesting work on replacement of serum free media using protein hydrolysates. The authors should perform experiments on the effect of protein hydrolysates on cell differentiation.
Why use zebrafish embryonic stem cell? Even though this stem cell is commercially available, the conditions established in the study may not be applicable to other types stem cells. Therefore, it will increase the value of the study to expand the work to other stem cells.
Protein hydrolysate production: not sure why used dried protein hydrolysate for calculate “yield”, but freeze-dried protein hydrolysate for “productivity”?
What is the rationale for use “alcalase”?
Cell culture: have the authors studied the effect of protein hydrolysates on cell differentiation?
There is a need to characterize the protein hydrolysates to understand the reason for the toxicity of these hydrolysates at high concentrations?
There are incomplete citations such as #26
Author Response
Interesting work on replacement of serum free media using protein hydrolysates. The authors should perform experiments on the effect of protein hydrolysates on cell differentiation.
- Thank you! We are also working on that, however, as you know, differentiation process is a long process, and before getting into that stage, we need to make sure that what concentration of protein hydrolysates are suitable for cell growth. The goal of this research is developing a low or serum free media.
Why use zebrafish embryonic stem cell? Even though this stem cell is commercially available, the conditions established in the study may not be applicable to other types stem cells. Therefore, it will increase the value of the study to expand the work to other stem cells.
- Correct, we applied Beefy9 and BeefyR which already have been used successfully for beef cell lines, for fish, and they killed our cell lines. Media for every species needs to be optimized. We selected Zebra fish since there is no other fibroblast or muscle line cell line available in the market from seafood. In addition, there is a startup company, using zebrafish cells to produce fish protein blocks like surimi.
Protein hydrolysate production: not sure why used dried protein hydrolysate for calculate “yield”, but freeze-dried protein hydrolysate for “productivity”?
- Thank you! They are the same, we freeze dried the protein hydrolysates.
What is the rationale for use “alcalase”?
- Since we screened nine protein sources, animals and plants, we decided to work with one enzyme and selected Alcalase since based on the literature and our experiment, it may provide a better hydrolysates in terms of peptide size.
Cell culture: have the authors studied the effect of protein hydrolysates on cell differentiation?
- Not yet, we are exploring it for the next step.
There is a need to characterize the protein hydrolysates to understand the reason for the toxicity of these hydrolysates at high concentrations?
- Protein hydrolysates have strong antioxidative and antiproliferation properties against cells at higher concentrations.
There are incomplete citations such as #26
- Thank you! We fixed them.
Reviewer 2 Report
The manuscript “biomolecules-1975897” describes a work aimed at evaluating protein hydrolysates (from Black soldier fly (BSF), cricket, 18 oyster, mussel, and lugworm) as potential serum-substitutes in cultured fish production. Overall, the paper is an adequate presentation of a carefully designed experiment. My (relatively) major concern is the luck of adequate control in the cell study where the performances of the different hydrolysates are measured. It is understandable that there is no gold standard of a control for protein hydrolysates, but, it would have been useful to compare the results with for example un-hydrolyzed protein (e.g., casein). This way one can attribute the effects form the hydrolysate to the special peptide, not just a protein/AA supplementation effect. I, therefore, suggest that the authors comment on how the findings will compare with other generic proteins and AAs. Nevertheless, I recommend the paper (with minor revision) for a publication in biomolecules.
Introduction:
The introduction is well organized except some imbalances in terms of the topics covered. For example, an introduction of such a work is expected to highlight the biotechnological process used (i.e., enzymatic protein hydrolysis (EPH)). The author should, to the least, define EPH and guide the reader to relevant references. At the same time more than necessary information was provided about a single parameter measured (i.e., LDH). This could be brief.
Line62-64: Please revise the sentence for a better readability.
Results and Discussion
Page 7/8: The authors needs to discuss correlation of hydrophobicity with bioactivity of peptides with more care. This will depend very much on the bioactivity (“therapeutic target protein”) and the specific sequence of the peptide. Therefore, the authors should revise the discussion and specifically the sentence in line 282-285).
Page 7: Give full name of OHC first and use abbreviation thereafter. Not the other way round.
Line 346-352: Please move most or all of this the Materials and Methods.
Author Response
The manuscript “biomolecules-1975897” describes a work aimed at evaluating protein hydrolysates (from Black soldier fly (BSF), cricket, 18 oyster, mussel, and lugworm) as potential serum-substitutes in cultured fish production. Overall, the paper is an adequate presentation of a carefully designed experiment. My (relatively) major concern is the luck of adequate control in the cell study where the performances of the different hydrolysates are measured. It is understandable that there is no gold standard of a control for protein hydrolysates, but, it would have been useful to compare the results with for example un-hydrolyzed protein (e.g., casein). This way one can attribute the effects form the hydrolysate to the special peptide, not just a protein/AA supplementation effect. I, therefore, suggest that the authors comment on how the findings will compare with other generic proteins and AAs. Nevertheless, I recommend the paper (with minor revision) for a publication in biomolecules.
- Thank you for the comment. We have thought about using standard protein, however, as you said, the standard protein should be from the same protein hydrolysates. In that case, we would have so much data, which the conclusion would be difficult. We decided to use protein isolates as another independent research.
Introduction:
The introduction is well organized except some imbalances in terms of the topics covered. For example, an introduction of such a work is expected to highlight the biotechnological process used (i.e., enzymatic protein hydrolysis (EPH)). The author should, to the least, define EPH and guide the reader to relevant references. At the same time more than necessary information was provided about a single parameter measured (i.e., LDH). This could be brief.
- Thank you! Revised.
Line62-64: Please revise the sentence for a better readability.
- Revised.
Results and Discussion
Page 7/8: The authors needs to discuss correlation of hydrophobicity with bioactivity of peptides with more care. This will depend very much on the bioactivity (“therapeutic target protein”) and the specific sequence of the peptide. Therefore, the authors should revise the discussion and specifically the sentence in line 282-285).
- Revised.
Page 7: Give full name of OHC first and use abbreviation thereafter. Not the other way round.
- Revised.
Line 346-352: Please move most or all of this the Materials and Methods.
- Revised.